# Low Antenatal Care Number of Consultations Is Associated with Gestational Weight Gain and Birth Weight of Offspring of Teenage Mothers: A Study Based on Colombian and Mexican Cohorts

**DOI:** 10.3390/nu16213726

**Published:** 2024-10-31

**Authors:** Reyna Sámano, Hugo Martínez-Rojano, Gabriela Chico-Barba, María Eugenia Mendoza-Flores, María Eugenia Flores-Quijano, Ricardo Gamboa, Andrea Luna-Hidalgo, Sandra L Restrepo-Mesa, Jennifer Mier-Cabrera, Guillermina Peña-Camacho

**Affiliations:** 1Coordinación de Nutrición y Bioprogramación, Instituto Nacional de Perinatología “Isidro Espinosa de los Reyes”, Mexico City 11000, Mexico; ssmr0119@yahoo.com.mx (R.S.); gabyc3@gmail.com (G.C.-B.); tina14mx@yahoo.com (M.E.M.-F.); alunahgo@gmail.com (A.L.-H.); jennifer.mier@gmail.com (J.M.-C.); 2Sección de Estudios de Posgrado e Investigación, Escuela Superior de Medicina, Instituto Politécnico Nacional, Mexico City 11340, Mexico; 3Programa de Maestría y Doctorado en Ciencias Médicas, Odontológicas y de la Salud, Universidad Nacional Autónoma de México, Mexico City 04510, Mexico; 4Departamento de Fisiología, Instituto Nacional de Cardiología “Ignacio Chávez”, Mexico City 14080, Mexico; rgamboaa_2000@yahoo.com; 5Centro Universitario Incarnate Word, Licenciatura en Nutrición, Campus Ciudad de México, Mexico City 03100, Mexico; 6Escuela de Nutrición y Dietética, Universidad de Antioquía, Cra. 75 #65-87, Pilarica, Medellín, Robledo, Medellín 050010, Antioquia, Colombia; sandra.restrepo@udea.edu.co; 7Departamento de Trabajo Social, Instituto Nacional de Perinatología “Isidro Espinosa de los Reyes”, Mexico City 11000, Mexico; guipuzcoas@hotmail.com

**Keywords:** pregnant teenagers, pregestational body mass index, pregnant teens, obesity, antenatal visits, teenager mothers

## Abstract

Background: More than 70% of pregnant adolescents in developing countries experience inappropriate gestational weight gain (GWG). Objective: To determine the association of the number of antenatal care visits (ANC) with GWG, birth weight, and their differences between two countries. Methods: A prospective study was conducted in two cohorts of adolescents, one from Mexico and one from Colombia. The study calculated pregestational body mass index (BMI), obtained GWG and birth weight, and collected socioeconomic characteristics. Birth weight was categorized according to gestational age. A total of 690 mother-child pairs were included, of which 42.6% were Colombian and 57.4% Mexican. Results: The study found no association between socioeconomic characteristics and GWG or birth weight. Colombian adolescents were more likely to experience insufficient GWG (68%), compared with 36% of Mexican adolescents. Colombian adolescents who attended fewer than eight ANC visits were at increased risk of insufficient GWG, whereas Mexican adolescents were at increased risk of excessive GWG. Mexican adolescents who began their pregnancies overweight or obese were at increased risk of excessive GWG. Fewer than eight ANC visits were associated with small for gestational age (SGA) in the Mexican cohort. Conclusions: Inadequate numbers of ANC visits were associated with excessive and insufficient GWG, and SGA. Promoting ANC in adolescent pregnancy is essential to prevent suboptimal GWG and SGA. This study highlights the need for interventions targeting pregnant adolescents from low socioeconomic backgrounds, prioritizing early initiation of prenatal care (first trimester) and a drastic reduction in the high rates of cesarean sections in this group.

## 1. Introduction

Data from 2023 on adolescent birth rates for ages 15–19, provided by the World Health Organization, show significant regional differences. The Americas have a rate of 39.5 births per 1000 women, Africa has the highest rate at 95.5, and Europe has the lowest at 12.7 births per 1000 women. According to the latest available information for individual countries in Latin America, Colombia (most recent data 2021) has 53.47 births per 1000 women, and Mexico (2019) has 50.67 births per 1000 women. These rates position them among the higher end within the region, where countries like Nicaragua have the highest rate at 80.64 (2021), and Bermuda has the lowest at 3.73 (2019). Among younger teenagers aged 10–14, the regional figure is 1.5 births per 1000 women, compared to 4.3 in Africa and 0.1 in Europe. In this age group, Mexico has a rate of 0.75 births per 1000 women, while Colombia has a rate of 2.28 births per 1000 women [1,2].

In certain contexts, early pregnancy and childbirth may be perceived as desirable means to fulfill traditional marriage expectations or attain adult status and social advancement. However predominantly, these pregnancies are unintended or unwanted, resulting from a myriad of factors. These include insufficient access to comprehensive reproductive health information, limited access to contraceptive methods, and exposure to sexual and gender-based violence [3,4].

Regardless of the underlying cause, teenager girls who experience pregnancy often share common characteristics: lower levels of formal education, residence in rural areas, and living in low-income households [5]. The onset of early motherhood, exacerbates pre-existing socioeconomic challenges, perpetuating a cycle of poverty and exclusion for both the teenager mother and her child. This cycle persists through diminished educational attainment, limited work opportunities, decreased labor force participation, and subsequently, lower income levels [3,6].

In addition to the socioeconomic disadvantages that teenager mothers face, pregnancy at this early stage poses significant health risks. These risks include low birth weight, premature birth, neonatal death, cephalopelvic disproportion, and even maternal death. However, there is an ongoing debate about whether these adverse outcomes are primarily due to biological factors—such as the physical immaturity and increased metabolic needs of young mothers—or the impoverished conditions in which many of these pregnancies occur [7,8]. Therefore, pregnancy during this critical growth phase further elevates these nutritional demands, posing a significant challenge, especially for girls from disadvantaged socioeconomic backgrounds who often begin pregnancy with inadequate nutritional stores. This scenario can lead to competition between the mother and the growing fetus, for dietary energy and nutrients [9,10,11]. Factors such as chronological age, the time since menarche, height, and pregestational BMI can influence the balance between growth and the nutritional needs of pregnancy, potentially increasing or reducing the risk of negative maternal and infant outcomes, including inadequate GWG, premature birth, and low birth weight [8,12,13].

On the other hand, socioeconomic factors such as household income level, family and social support, as well as prenatal care have also been associated with inadequate GWG and adverse birth weight outcomes [12,13,14,15]. Although substantial evidence highlights the importance of clinical, social, and economic factors in shaping teenager health outcomes and their offspring, further research is needed to understand how these factors influence gestational weight and birth outcomes. It is known that prenatal care is crucial for preventing adverse outcomes; however, the impact of prenatal care on these outcomes in teenagers is not fully understood, particularly given that the initiation of prenatal care in this population is often delayed.

Antenatal care (ANC) is essential for improving perinatal outcomes. The World Health Organization (WHO) recommends covering at least eight consultations [16]. Delayed antenatal care is associated with late pregnancy identification and poor knowledge of care options [17]. Perinatal outcomes have been reported in adult women [18,19], and qualitative research has identified reasons for late initiation of prenatal care in adult women [20,21,22]. Nevertheless, the association between late ANC or the number of ANC visits and perinatal outcomes of teenage pregnancy remains uncertain. In Latin American countries accurate information of ANC is lacking. Despite a similar prevalence of teenage pregnancy in Mexico and Colombia [1], there is a lack of information on GWG and birth weight outcomes in relation to the number of antenatal care visits [23]. Most of the available data is in adult women [22]. Unlike adult women, teenagers are still undergoing physical growth, which may affect their GWG, and the prevalence of small for gestational age (SGA) infants is higher in this population. Therefore, the aim of this study is to investigate the association between ANC consultations, GWG, and perinatal outcomes, while comparing differences between two countries with a similar prevalence of teenage pregnancy [24].

## 2. Materials and Methods

### 2.1. Study Population

Six hundred ninety teenagers with single pregnancies and no previous births, aged between 11 and 19 years, living in urban areas of Medellin, Colombia, and in Mexico City, Mexico, including neighboring conurbated areas, were enrolled in this analytical prospective study.

In Colombia, participants were recruited from the antenatal control program of the Medellin Public Hospital Network and various clinics, using a stratified, proportional, and representative sampling approach in each healthcare center. In Mexico, teenagers receiving prenatal care at the Instituto Nacional de Perinatología, were recruited from the outpatient area using a non-probabilistic sampling method, trained personnel invited all teenagers to participate in the study. Follow-up in Colombia consisted of monthly nutritional visits, while in Mexico it was quarterly. In the consultations Anthropometric body weight and dietetic measurements were performed.

Due to the potential interaction with pregnancy weight gain and newborn outcomes, we excluded teenagers with a history of chronic infectious diseases, chronic diseases such as diabetes, systemic lupus erythematosus, and rheumatoid arthritis, as well as those with substance use disorders, including smoking, alcohol use, and drug use. Informed consent and assent were obtained from all participating teenagers and their parents/guardians in both countries.

### 2.2. Sociodemographic and Gyneco-Obstetric Information

Three authors of the present research applied a questionnaire to gather data on sociodemographic variables, including chronological age, years of education completed, marital status, parental family structure, and number of household members before pregnancy. Additionally, information was gathered on household welfare and financial support for personal expenses during pregnancy.

Educational lag was defined as a teenager’s current grade or highest completed grade being two or more years behind the expected level for their age. Marital status was classified according to the United Nations Children’s Fund (UNICEF) definition of child marriage, categorizing participants as single, married, or living with a partner [25]. However, in compliance with Mexico’s Federal Civil Code, which mandates a minimum age of 18 for marriage [26], we combined the categories of ‘married’ and ‘cohabiting’ into a single category labelled ‘lives with her baby’s father’, asking whether they lived with the baby’s father before and during pregnancy. The parental family type was categorized as nuclear, extended, or uniparental. Household socioeconomic levels were assessed using the AMAI Rule 8X7, a methodology developed by the Mexican Association of Market Intelligence and Public Opinion Agencies (AMAI) that classifies households into seven socioeconomic status based on the head of the household’s capacity to meet its members’ needs [27]. In Colombia, a family wellbeing scale was used, with parameters for classifying households into six socioeconomic status levels, with level 1 being the lowest and level 6 being the highest [28]. In both cohorts, the categories were synthesized and standardized into three socioeconomic strata: middle, low-middle, and low.

Furthermore, personal reproductive history, as well as information about their pregnancy, was collected. This included age at menarche, duration between the onset of menarche and the time of pregnancy, method of delivery (vaginal delivery, and cesarean section), and week of gestation at the first prenatal care visit. For the analysis, this variable was categorized into trimesters, the first trimester spanned from the start of gestation to 14 weeks and 0 days, the second trimester from 14 weeks and 1 day to 28 weeks and 0 days, and the third trimester from 28 weeks and 1 day until delivery.

We counted the number of medical ANC consultations that teenagers had during their pregnancy. We also recorded the gestational week at which each visit took place. We categorized this information, according to the criteria proposed by the WHO regarding the ideal number and timing of appointments: “a minimum of eight contacts are recommended to reduce perinatal mortality and improve women’s experience of care”. Based on this criteria, having less than eight ANC consultations was considered inadequate [29].

We gathered data on the newborn’s sex and gestational age (GA). GA at birth was calculated by determining the number of completed weeks of gestation based on the estimated delivery date from clinical records. This estimation can rely on either the mother’s last normal menstrual period or the Capurro method, a clinical assessment tool for newborn gestational age. If the gestational age was less than 37 weeks, it was classified as preterm, while a GA between 37 and 42 weeks was identified as a term delivery.

### 2.3. Anthropometric Variables

Trained nutritionists obtained the following information. Participants self-reported their pregestational weight at the study visit. Participants were weighed to obtain pre-birth maternal weight using a Tanita digital scale (model BWB-800, Tokyo, Japan) with an accuracy of 0.1 kg, and height was measured with an accuracy of 0.1 cm using a SECA stadiometer (model 208, Hamburg, Germany). This information was used to calculate pre-pregnancy BMI by dividing maternal weight in kilograms by the square height in meters (kg/m²). The calculated BMI was then categorized into percentiles using the AnthroPlus^®^ program, developed by the World Health Organization in Geneva, Switzerland. This classification system stratified BMI values into specific percentile ranges: underweight (<3rd percentile), normal weight (3rd–85th percentile), overweight (86th–97th percentile), and obesity (>97th percentile) [30].

Total gestational weight gain was determined by subtracting the pregestational weight (in kilograms) from the last weight observed during the study visit, typically occurring 1–2 weeks before delivery. In cases of premature birth, data were retrieved from clinical records. The recommended gestational weight gain was calculated according to the Institute of Medicine’s (IOM) guidelines [12] using the equation:Recommended GWG (kg) at the last observed weight measure = recommended GWG first trimester (kg) + ((GA − 13.86) × (recommended weekly GWG in second and third-trimesters)).

The values used in the equation are as follows: The recommended first-trimester GWG is 2 kg (for underweight and normal pre-pregnancy BMI), 1 kg (for overweight), and 0.50 kg (for obesity). For the last two trimesters, the weekly GWG recommended rate is 0.51 kg (for underweight), 0.42 kg (for normal), 0.28 kg (for overweight), and 0.22 kg (for obesity) [12]. The gestational age at the time of the last visit was recorded.

Using the total GWG and the recommended GWG (kg) at the last observed weight, we calculated the percentage of recommended GWG achieved, using the following equation:% Adequacy GWG = (Observed GWG ÷ Recommended GWG) × 100

Finally, the percentage of recommended GWG achieved was categorized as insufficient (<90%), adequate (90–125%), and excessive (>125%) [31]. Both excessive and insufficient GWG were considered inadequate.

Anthropometric data for the newborn (NB), including birth weight and length, were measured within the first hour after birth. Weight was measured in grams (g) to the nearest 0.1 g using a calibrated scale (SECA 374, Hamburg, Germany, model “Baby and Mommy”), while length was measured in centimeters to the nearest 0.1 cm using an infantometer (SECA 416, Hamburg, Germany). The NB’s birth weight was categorized using the Intergrowth criteria: small for gestational age (SGA) for <10th percentile, adequate for gestational age (AGA) for 10th–90th percentiles, and large for gestational age (LGA) for >90th percentile [32]. Additionally, the Z-score of the NB’s weight was calculated using the WHO 2006 growth charts [30]. All anthropometric measures were performed by Lohman technique [33].

### 2.4. Ethical Considerations

The study complied with the Declaration of Helsinki regarding research involving human subjects. Approval for the portion conducted in Mexico was obtained from the Medical Ethics Committee of the National Institute of Perinatology, Ministry of Health of Mexico, registration number: INPer 2017-2-101. In Colombia, approval was granted by the Bioethics Committee of the Faculty of Dentistry of the Universidad de Antioquia, and the research division of the University of Antioquia in Medellín, Colombia. As previously mentioned, teenager participants and their parents or guardians were asked to read and sign the informed consent letter for parents and the assent letter for teenagers. The hospital institutions where the studies were conducted provided their endorsement and authorization for reviewing newborns’ medical records. All data collection was confidential, adhering to ethical principles such as autonomy and safety.

### 2.5. Statistical Analysis

Descriptive statistics were conducted for all variables based on GWG categories. Continuous variables were presented using mean and standard deviation, while categorical variables were represented by percentages. The distribution of continuous variables was assessed using the Kolmogorov-Smirnov test. Bivariate analyses were conducted with continuous variables using the Student’s *t*-test or Mann-Whitney U test, and categorical variables using the Chi-square test.

To comprehend the impact of factors such as maternal age, socioeconomic status, and education level on the probability of insufficient and excessive gestational weight gain, as well as small for gestational age (SGA), within each teenager group, we conducted a series of logistic regression analyses, which allowed us to calculate odds ratios (ORs) along with their corresponding 95% confidence intervals (CIs).

The first and second series each included three models (M) assessing the association between excessive and insufficient GWG: M1 was unadjusted, M2 was adjusted for age, and M3 was adjusted for age, socioeconomic status, education level, and gynecological age. The third series assessed the factors associated with SGA. The models were the same, except for M3, which also controlled for gestational weight gain. The models included potential confounding variables, such as age, socioeconomic status, education level, and gynecological age. The analyses were conducted using IBM SPSS Statistics version 21 for Windows (IBM Corp., North Castle, NY, USA). Statistical significance was defined as *p* < 0.05.

## 3. Results

In total, there were 690 participating dyads, of which 294 (42.6%) were Colombian and 396 (57.4%) were Mexican. In Table 1 and Table 2, we show the sociodemographic and general characteristics of both groups. The Colombian teenagers showed a higher average chronological age. Specifically, the proportion of teenagers in the age group of ≤15 years was 15.6% among the Colombian sample and 46.9% among the Mexican sample. This difference in age groups resulted in higher gynecological age and stature among the Colombian participants. Educational lag, defined as the discrepancy between a student’s age and their expected grade level, was three times more prevalent among Colombian teenagers (76.5% vs. 29%).

While a higher proportion of Mexican teenagers lived with their baby’s father before pregnancy (*p* = 0.005), this difference disappeared during pregnancy, with nearly 60% of both groups living with their partners (*p* = 0.557). In terms of family structure, extended families, with multiple generations living together, were more prevalent among Colombian women, which explains why there are more household members living together. In contrast, nuclear families, consisting primarily of parents and children, were more common among Mexicans.

Colombian teenagers generally resided in lower-income households compared to their Mexican counterparts, with a larger proportion of Colombian teenagers reporting annual household incomes below specific income threshold compared to Mexican teenagers. In both countries, parents or siblings were the primary sources of financial support for pregnant teenagers. Interestingly, financial support from partners was significantly more common among Colombian teenagers (50%) compared to Mexican teenagers (11%) (Table 1).

Regarding pBMI, 9% of Colombian teenagers had overweight/obesity before pregnancy, compared to 20% in the Mexican cohort (*p* ≤ 0.001). Regarding the timing of their first prenatal care visit, Colombian teenagers had their first contact with healthcare services earlier than their Mexican counterparts (*p* ≤ 0.001). However, significantly fewer Colombian teenagers than Mexican teenagers had the recommended eight or more ANC visits (see Table 2).

In terms of GWG, Colombian teenagers exhibited lower maximum weight gain compared to their Mexican counterparts (9 kg vs. 12 kg, *p* ≤ 0.001). According to the IOM weight gain classification (see Table 2), 68% of Colombian teenagers experienced insufficient weight gain, while only 12% had excessive gain. In contrast, Mexican teenagers showed a more varied distribution. 36% experienced insufficient weight gain, while 33% had excessive weight gain.

Colombian teenagers delivered babies with greater birth lengths and weights compared to Mexican teenagers. There was a trend toward a higher proportion of SGA infants among Mexican teenagers (17% vs. 24%, *p* = 0.069). The proportion of LGA infants was very low in both groups. No significant differences were observed in the mean gestational age at delivery between the study groups. However, Mexican teenagers had a higher percentage of preterm deliveries (*p* = 0.013). Additionally, a significantly higher proportion of deliveries among Mexican teenagers were via cesarean section.

Bivariate analysis (see Appendix A) revealed no associations between any of the outcome variables—GWG, SGA, prematurity, or cesarean section delivery—and the studied sociodemographic variables, regardless of the participant’s country of origin. However, other associations were found between number of antenatal care consultations and GWG and SGA.

In the Mexican cohort, excessive GWG, as defined by the IOM guidelines, was associated with starting pregestational overweight and obesity (*p* ≤ 0.001); having less than eight prenatal care contacts (*p* = 0.041) and a higher frequency of preterm birth (*p* = 0.040). Furthermore, SGA newborns were more likely to be born to mothers who were overweight or obese before pregnancy (*p* = 0.028), had less than eight prenatal care contacts (*p* = 0.006), and had a shorter gynecological age (*p* = 0.019).

Among Colombian women, attending less than eight prenatal care visits was associated with a higher frequency of insufficient weight gain among teenagers (*p* = 0.007) and a higher likelihood of delivering a SGA newborn (*p* = 0.045).

The adjusted logistic regression models revealed the following findings:

### Insufficient Gestational Weight Gain

Among Colombian teenagers, insufficient GWG was over five times more likely among those who attended seven or fewer prenatal care visits (OR 5.28, 95% CI 1.94–14.41). Conversely, insufficient GWG was significantly less likely among those who began pregnancy with pregestational overweight or obesity (OR 0.30, 95% CI 0.10–0.92).

In the Mexican sample, excessive GWG was more than twice as likely in those who attended seven or fewer prenatal care visits (OR 2.29, 95% CI 1.29–5.60) and more than six times as likely among those who started pregnancy with pregestational overweight or obesity (OR 6.06, 95% CI 3.44–10.66). Conversely, insufficient GWG was approximately 60% less likely among teenagers who began pregnancy with a high pre-pregnancy body mass index (OR 0.37, 95% CI 0.19–0.70) as shown in Table 3.

Data from the Mexican sample revealed that having less than the recommended number of prenatal care visits was significantly associated with an increased risk of delivering a small-for-gestational-age newborn among teenagers (Table 4).

## 4. Discussion

Despite attempts to promote and implement adequate antenatal care, the present study reports differences in the two countries regarding GWG and birth weight when teenagers had fewer than eight antenatal care visits. We highlight that less than 20% of our teenager participants attended an adequate number of ANC consultations. This finding differentially affected inadequate GWG for Colombian and Mexican teenagers but had a similar impact on SGA rates.

In terms of anthropometric variables, specifically the prevalence of overweight and obesity before pregnancy among our study participants, our findings reveal a significant discrepancy when compared to the most recent national data on these conditions in teenager females. In Colombia, the prevalence of overweight and obesity is reported at 21.1% [34], whereas in Mexico it is 41.0% [35]. Our study samples show a significantly lower prevalence of overweight and obesity compared to these national statistics.

Regarding GWG, Colombian teenagers who were older and had a lower pBMI experienced insufficient GWG. In contrast, younger and heavier Mexican teenagers had excessive weight gain. This disparity can be explained by three key factors: First, higher pre-pregnancy BMI is known to correlate with increased risk of excessive weight gain during pregnancy, independent of age [36], highlighting the influence of initial BMI on GWG tendency.

Second, the 2009 guidelines for pregnancy weight gain, which are still in use, recommend classifying teenagers using the World Health Organization (WHO) BMI cutoff points for adults. This classification system may lead to some teenagers being categorized in a lower BMI range, which could potentially result in higher recommended weight gain targets. Additionally, these guidelines suggest that “younger teenagers often need to gain more to improve birth outcomes” [12]. This recommendation has been integrated into other national guidelines, such as the 2016 Official Mexican Standard for the Care of Women During Pregnancy, Childbirth, and the Postpartum Period, and for Newborns [37], which advises teenagers to aim for the upper end of the GWG range. Such guidelines may increase the likelihood of higher weight gain recommendations for pregnant teenagers [38,39]. However, we found no previous research documenting this practice directly.

Third, studies have noted that teenagers in earlier stages of growth tend to gain more weight compared to those who have completed their growth [40,41]. Research conducted in a low socioeconomic city in the United States found that teenagers who were still growing accumulated more body fat by the end of pregnancy, whereas those who had completed growth showed similar weight and body composition to adult women [41]. Although our study did not directly assess the growth status of Colombian teenagers, given their higher gynecological age, it is likely that some had completed their growth. Typically, girls complete their growth approximately 2.5 to 3 years after menarche [42,43]. Therefore, the observed differences in GWG between Mexican and Colombian teenagers may be attributed to variations in initial BMI and growth status.

Regarding the moment antenatal care is initiated and the number of consultations, our study demonstrated that less than 20% of teenagers had seven or fewer ANC consultations. Previous studies have documented that teenagers are more likely to experience a lack of prenatal care, initiate their first prenatal consultation later than the first trimester, and have fewer prenatal visits overall [44,45,46,47]. For example, a large cohort study in Mexico found that the adjusted probability of adequate ANC consultations increased with age, with a frequency of 56% for the youngest group (10 to 14 years) compared to 65.5% among the young adult group (20 to 24 years) [44]. Other research conducted in Bosnia and Herzegovina found that 60% of teenager women received five or fewer ANC examinations, with none receiving more than eight consultations. In contrast, nearly 82% of adult women attended at least six ANC consultations [45]. Similarly, a study from South Africa reported that 43% of teenagers had fewer than four ANC consultations [46]. Several factors contribute to the lower frequency of ANC consultations among teenagers. These factors include the teenager’s and her partner’s level of education, socioeconomic status, parity, family structure, the desirability of the pregnancy, and personal barriers related to the teenager woman’s circumstances such as socioeconomic, and physical adverse conditions [13,14,44,46,47,48]. Additionally, exposure to mass media has been identified as a potential influencing factor [47].

Among Mexican teenagers, SGA newborns were more frequent than in the Colombian sample; this is a concern because it implies longer hospital length of stay and increased risk of morbidity, which could be associated with prematurity, as has been reported higher compared to adult women in two Mexican samples [36,49]. One explanation could be that young teenagers and their babies may experience competition for nutrients [42,43], potentially affecting their physical growth and leading to inadequate birth weight. This, coupled with low antenatal care visits, increases the risk of adverse perinatal outcomes [40].

Regarding the bivariate associations and multivariate models for GWG and SGA, none of the sociodemographic variables demonstrated association. Although previous studies have documented associations between sociodemographic factors and pregnancy outcomes, our findings did not reflect such relationships [50,51,52]. Despite differences between the teenager groups in each country, such as mean age, educational attainment, and cohabitation status, both groups shared a predominance of low socioeconomic status. This socioeconomic homogeneity might explain the absence of significant associations between GWG or SGA and the sociodemographic variables examined in our study.

The factors contributing to insufficient or excessive GWG among Mexican and Colombian teenagers were consistent across both groups. These factors include pre-pregnancy BMI and the number of antenatal care consultations received.

On the other hand, starting pregnancy with overweight or obesity has been shown to reduce the likelihood of insufficient weight gain among teenagers in both countries. For Mexican teenagers, this condition was also associated with a higher frequency of excessive GWG. Our study used WHO BMI cutoff points for age and sex to classify pregestational BMI, which prevented inaccuracies associated with IOM cutoff points designed for adults. IOM points tend to underestimate the proportion of women with pregestational overweight and obesity, often recommending higher weight gain for those misclassified as having a lower pre-pregnancy BMI (pBMI) [39]. By using WHO BMI cutoffs for age and sex, a more accurate classification of teenagers was achieved. Since the recommended weight gain range for those who start pregnancy with overweight or obesity is lower, it may be inadvertently difficult to gain below and easy to gain above the recommendation. This observation has been documented in previous studies [36].

We reported that Mexican teenagers began antenatal care later than the Colombian teenagers, nonetheless, both samples had less than the recommended number of prenatal care visits. This situation could potentially explain the higher frequency of excessive GWG, as teenagers may not have received adequate counseling regarding the importance of gestational weight gain and birth weight during this period of life [53,54].

On the other hand, a low number of antenatal care consultations has been linked to the development of various adverse pregnancy outcomes, such as low birth weight, prematurity, and maternal and infant morbidity and mortality [54]. Adequate antenatal care consultations are associated with improved birth outcomes, particularly for socially disadvantaged women, including teenagers, and can help reduce the disparity in adverse neonatal outcomes between teenager and adult women [44]. Specifically, adequate ANC was associated with a lower likelihood of having a SGA baby. Teenagers aged 10 to 14 years with proper prenatal follow-up had a 7.3% probability of delivering a low-birth-weight infant, compared to a 9.4% probability for those without such care. For instance, our study confirms that women who attended less than eight ANC visits were more likely to deliver SGA infants, particularly among younger teenagers compared to their counterparts. One possible explanation is that they sought ANC later and, therefore, probably did not receive adequate nutritional and medical advice during pregnancy [40]. However, this may compromise the growth potential of their offspring during early childhood [55] and negatively affect metabolic programming [56,57].

In some regions of Latin America, nutritional counseling is not included in antenatal care visits. This could be a contributing factor to the findings that 33% of Mexican teenagers experienced excessive gestational weight gain (GWG), 36% experienced insufficient GWG, and 68% of Colombian teenagers experienced insufficient GWG in our research. Health professionals should consider that many women are motivated to improve their health behaviors during pregnancy. Therefore, this period is often considered ideal for promoting healthy nutrition and physical activity to achieve appropriate gestational weight gain [58].

Our results highlight the importance of the number of antenatal care consultations among pregnant teenagers. Despite WHO recommendations [29], we lack information on the guidance provided by healthcare personnel regarding GWG and birth weight during these consultations. Addressing this gap is crucial for improving outcomes related to GWG and SGA, especially considering the tendency of teenagers to begin antenatal care late.

In our study, the rate of cesarean sections in both groups exceeded the ideal threshold of 15% [59]. This was expected, given that both countries are part of a region with some of the highest cesarean section rates globally [59]. The most recent data for each country indicates a rate of 45.8% in Colombia (2016) and 40.7% in Mexico (2013–2015) [60]. Notably, the frequency of cesarean sections among Mexican teenagers was more than double the rate, aligning closely with the national figure, whereas among Colombian teenagers, the frequency was less than half the national rate.

Teenage pregnancy has consistently been associated with a lower incidence of cesarean births in studies conducted across various regions of the world [7,61,62,63]. However, a study found no difference between teenagers and adults [64]. Other research reports a higher incidence of cesarean births among teenagers with specific characteristics, such as those aged 15 or younger with a diagnosis of “presumed” cephalopelvic disproportion [7], or those with a higher pre-pregnancy BMI and excessive gestational weight gain [65]. Additionally, some studies indicate that teenagers generally face a higher risk of cesarean delivery compared to adults [66,67]. The Mexican sample had a higher frequency of overweight and obesity pregestational BMI categories, though this was not associated with cesarean delivery. However, it may be linked to a low number of antenatal care visits, as reported by Ebeigbe [67]. This could be explained by a lack of follow-up [68] and inadequate information provided to identify potential complications during pregnancy.

In teenagers younger than 15 years old, the fetus may be in an unfavorable position, and the mother’s pelvis may not be fully developed, making vaginal delivery more challenging. Mexican teenagers, due to their younger age and lower number of antenatal care visits, may have a narrower pelvis than older Colombian women, as demonstrated by studies by MacDonald JA [68] and Marino JL et al. [69].

The institutions participating in our study in Colombia and Mexico provide medical assistance, particularly perinatal care, to individuals without social security who live in the lowest socioeconomic strata of cities and surrounding urban areas. While this focus is limited, as the findings may not apply to pregnant teenagers from different socioeconomic backgrounds, it offers a distinct advantage. As teenage pregnancy is predominantly concentrated within the lower socioeconomic strata in Latin America [70], our study’s focus on this population is particularly relevant to the region’s most affected groups. The latest available data indicate that around 27.1% and 22.6% of teenagers in the very low socioeconomic category from Colombia and Mexico, respectively, have experienced pregnancy, compared to just 7.2% and 4.4% in the highest socioeconomic categories [71,72]. In our study, women from both countries were categorized under low socioeconomic status.

One difference between the samples from the two countries is the higher chronological and gynecological age observed among the women in the Colombian group. It is noteworthy that a larger proportion of Colombian teenagers, who on average were older, exhibited greater educational lag compared to the younger Mexican teenagers. Since the women in the study are first-time mothers and already face educational lag, it suggests that some of these teenagers had stopped studying before becoming pregnant. This supports the hypothesis that attending school is a protective factor against teenager a fertility [73]. Furthermore, it can be argued that cultural and socioeconomic contexts, often characterized by scarcity, lack of motivation, and minimal family support for continuing education, contribute to teenagers abandoning their studies early. In such environments, these young women do not perceive any benefit in postponing motherhood, and some of them may be placed in traditional gender roles where pregnancy is accepted and expected [74,75].

One interesting observation is the high proportion of teenagers who were living with their baby’s father prior to becoming pregnant. This is particularly striking given that, according to the legal framework in some countries, such as Mexico, marriage among minors is discouraged [26]. Despite this, a significant number of young women live with their partners before pregnancy, and this proportion increases after pregnancy occurs. Research has shown that living with a partner before pregnancy significantly raises the likelihood of becoming pregnant. Data from a national Mexican survey indicates that teenagers who live with a partner are nine times more likely to experience pregnancy compared to those who do not [73].

### 4.1. This Study Has Several Limitations

Retrospective design: The retrospective nature of the study may have resulted in misclassification of some births due to limitations in the available data.

Self-reported weight: Pregestational weight was self-reported, potentially introducing bias. However, evidence supports the use of self-reported weight as a practical and cost-effective measure [76].

Incomplete weight gain data: The calculation of GWG was based on total weight gain during pregnancy, lacking information on the rate and timing of weight gain. This limits our understanding of the weight gain trajectory.

Limited generalizability: Although our study sample comprised mostly low socioeconomic status teenagers lacking health insurance from the Public Hospital Network of Medellín, Colombia, and the Instituto Nacional de Perinatología, Mexico, the findings may not be generalizable to all pregnant teenagers in Colombia, Mexico, and other Latin American countries due to the study’s non-representative sample.

Inaccuracy of IOM 2009 recommendations: This study utilized the 2009 IOM recommendations, which are appropriate for normal-weight and overweight women but inaccurate for underweight and morbidly obese women. However, the low proportion of underweight (3–4%) and obese (2–3%) women in the sample minimizes the impact of this limitation [77].

Questionable applicability of Intergrowth-21 curves: Following international recommendations, this study used the Intergrowth-21 curves. However, we found 17% low birth weight (LBW) in Colombian teenagers and 24% in Mexican teenagers, compared to the expected 10%. This suggests that Intergrowth-21 may be inadequate for these populations, a concern echoed in numerous international reports from diverse ethnic groups [78,79,80,81,82,83].

Relevance despite limitations: Despite these limitations, the findings align with the higher prevalence of teenage pregnancy reported among low socioeconomic groups in both countries [84,85]. This suggests that our findings may be relevant to a larger population of pregnant teenagers in Colombia, Mexico, and other Latin American countries with similar socioeconomic characteristics.

### 4.2. Future Research Directions

Future research is needed to address these limitations and provide further insights into teenager pregnancy outcomes. Specifically:

Further studies with more representative samples should be conducted to confirm the findings.

Future studies should include data on the rate and timing of weight gain during pregnancy to provide a more comprehensive understanding of GWG trajectories.

Research is needed to better understand the reasons for late antenatal care visits, and these studies should include information about the guidance provided to teenagers by health personnel on GWG and birth weight.

### 4.3. Key Findings

Despite these limitations, our study highlights the importance of antenatal care consultations for pregnant teenagers. It emphasizes the need to address factors contributing to late antenatal care initiation and to provide comprehensive guidance on gestational weight gain and birth weight to teenagers.

Latin America, particularly Mexico, reports the highest global rates of Cesarean sections in first-time mothers [86,87]. Therefore, urgent public health interventions are needed to reduce medical interventions during teenage childbirth, as this creates an inevitable domino effect on subsequent deliveries [88].

## 5. Conclusions

Our study’s findings are crucial for understanding the impact of antenatal care visits on teenage pregnancies. We found a higher frequency of excessive gestational weight gain (GWG) among Mexican teenagers and insufficient GWG among Colombian teenagers, both associated with the number of antenatal care consultations. Small for gestational age (SGA) was also associated with inadequate antenatal care, particularly in the Mexican sample.

This study highlights the need for targeted interventions focused on pregnant teenagers from low socioeconomic backgrounds. Prioritizing early prenatal care initiation (first trimester) and drastically reducing high Cesarean section rates in this group are crucial.

It is crucial that Mexican healthcare workers strive to significantly reduce the percentage of teenagers who begin prenatal care in the second trimester of pregnancy; this currently stands at 64%.

These results highlight the need for further research to better understand and address this issue.

## Figures and Tables

**Table 1 nutrients-16-03726-t001:** Socioeconomic characteristics of Colombian and Mexican pregnant teenagers (N = 690), frequency (%).

Variables	Colombia, *n* = 294	Mexico, *n* = 396	*p*
Age (years) ^a^	17.2 ± 1	15.4 ± 1	≤0.001
Educational lag (%) ≥ 2 years ^b^	225 (76.5)	115 (29)	≤0.001
Lives with the baby’s father (before pregnancy) (%) ^b^	120 (26)	175 (44)	0.005
Lives with the baby’s father (during pregnancy) (%) ^b^	173 (59)	222 (56)	0.557
Family type before pregnancy (%) ^b^			
Nuclear	85 (29)	242 (61)	≤0.001
Extended	147 (50)	95 (24)	
Uniparental	62 (21)	59 (15)	
People living in your house (*n*)	5 (4–6)	3 (2–5)	≤0.001
Household welfare (%) ^b^			
Middle-Low	59 (20	107 (27)	≤0.001
Low	132 (45)	238 (60)	
Very low	103 (35)	51 (13)	
Financial support for personal expenses ^b^			
Herself	15 (5)	0 (0)	≤0.001
Partner	141 (48)	44 (11)	≤0.001
Parents/sibling	188 (64)	333 (84)	≤0.001
Extended family	44 (15)	20 (5)	≤0.001

^a^: Data expressed as mean ± standard deviation *p* value by Student-*t* test, ^b^: *p*-value by Pearson’s Chi-Square.

**Table 2 nutrients-16-03726-t002:** Gyneco-obstetric and anthropometric variables for Colombian and Mexican dyads (N = 690).

Variables	Colombia, *n* = 294	Mexico, *n* = 396	*p*
Menarche (years) ^a^	12 (12–13)	12 (11–12)	≤0.001
Gynecologic age (years)	5 (4–6)	4 (3–5)	≤0.001
≥4 years after menarche (%) ^c^	223 (76)	225 (57)	≤0.001
≤3 years after menarche (%) ^c^	71 (24)	171 (43)	
Height (cm) ^a^	156.3 ± 5	155.5 ± 5	0.046
Pregestational weight (kg)	50.8 ± 8	51.7 ± 8	0.175
pBMI ^a^	20.8 ± 2	21.5 ± 3	0.012
pBMI (Z-Score) ^b^	0.2 (−0.8, 0.3)	0.3 (−0.6, 0.9)	≤0.001
pBMI classification (*n*, %) ^c^			
Underweight	9 (3)	16 (4)	≤0.001
Normal	259 (88)	301 (76)	
Overweight	20 (7)	67 (17)	
Obesity	6 (2)	12 (3)	
GWG Maximum gestational weight (kg) ^a^	60.5 ± 10	64.2 ± 9	≤0.001
Total GWG (kg) ^b^	9 (5.8–11.6)	12 (9.5–15)	≤0.001
GWG adequacy (%) ^b^	71 (47–100)	103 (80–136)	≤0.001
GWG classification (*n*, %) ^c^			
Insufficient	200 (68)	143 (36)	≤0.001
Adequate	59 (20)	123 (31)	
Excessive	35 (12)	130 (33)	
Antenatal Care			
Initiation of antenatal care (weeks) ^b^	13 (10–18)	19 (15–23)	≤0.001
Number of antenatal consultations ^b^			
≥8 ^c^ adequate	33 (11)	65 (16)	0.039
≤7 ^c^ inadequate	263 (89)	331 (84)	
Initiation of antenatal care by trimester (%) ^c^			
First	164 (56)	79 (20)	≤0.001
Second	103 (35)	254 (64)	
Third	27 (9)	63 (16)	
Delivery (%) ^c^			
Vaginal birth	229 (78)	198 (50)	≤0.001
C-section	65 (22)	198 (50)	
Newborns			
Sex (%) ^c^			0.123
Girls	138 (47)	206 (52)	
Boys	156 (53)	190 (48)	
Weight (g) ^b^	3125 (2812–3360)	2924 (2655–3204)	≤0.001
Length (cm) ^b^	49.7 (48–51)	49 (47–50)	≤0.001
Weight (Z-score) ^b^	−0.4 (−0.9, 0.09)	−0.7 (−1.2, −0.07)	≤0.001
Newborns birth weight (%) ^c^			
SGA	50 (17)	95 (24)	0.069
AGA	238 (81)	297 (75)	
LGA	6 (2)	4 (1)	
Gestational age			
Gestational age at birth (weeks) ^b^	39 (38–40)	39 (38–39)	0.076
Term (%) ^c^	279 (95)	356 (90)	0.013
Pre-term (%) ^c^	15 (5)	40 (10)	

pBMI: Pregestational body mass index; SGA: small for gestational age; AGA: adequate for gestational age; LGA: large for gestational age, ^a^: Data expressed as mean ± standard deviation, *p*-value by Student’s *t*-test. ^b^: Data expressed as median (25 percentile–75 percentile), *p*-value by Mann-Whitney U test. ^c^: Data expressed as frequency (%), *p*-value determined by Pearson’s Chi-Square test.

**Table 3 nutrients-16-03726-t003:** Predictor variables of inadequate gestational weight gain among pregnant teenagers in Colombia and Mexico.

	Colombia, *n* =294				Mexico, *n* = 396			
	Insufficient		Excessive		Insufficient		Excessive	
	OR	95% CI	OR	95% CI	OR	95% CI	OR	95% CI
≤7 antenatal care visits								
M1	3.99	1.61–9.89	0.43	0.14–1.32	0.85	0.48–1.50	2.53	1.25–5.08
M2	4.26	1.68–10.68	0.37	0.16–1.17	0.83	0.46–1.48	2.55	1.27–5.13
M3	5.28	1.94–14.41	0.33	0.39–1.52	0.85	0.46–1.58	2.29	1.29–5.60
Married/Cohabiting								
M1	1.88	0.97–3.65	0.63	0.24–1.64	0.80	0.51–1.24	1.28	0.83–2.00
M2	1.88	0.97–3.68	0.63	0.23–1.62	0.80	0.52–1.25	1.30	0.83–2.02
M3	1.83	0.92–3.65	0.59	0.22–3.69	0.81	0.51–1.27	1.35	0.86–2.11
Without partner support								
M1	0.61	0.32–1.16	2.14	0.83–5.54	0.65	0.32–1.30	0.96	0.46–1.95
M2	0.61	0.32–2.66	2.15	0.93–5.56	0.65	0.33–1.31	0.96	0.47–1.97
M3	0.65	0.33–1.25	2.31	0.86–6.21	0.66	0.32–1.34	0.91	0.44–1.89
Without parents support								
M1	1.64	0.83–3.25	1.07	0.43–2.71	1.66	0.92–3.00	0.86	0.46–1.60
M2	1.65	0.83–3.26	1.07	0.42–2.71	1.66	0.92–2.98	0.85	0.45–1.58
M3	1.53	0.76–3.09	1.12	0.43–1.92	1.69	0.94–3.09	0.88	0.47–1.66
pBMI overweight/obesity								
M1	0.38	0.13–1.10	2.97	0.85–10.31	0.38	0.20–0.72	5.81	3.32–10.14
M2	0.37	0.13–1.09	2.97	0.85–10.36	0.38	0.20–0.71	5.88	3.36–10.31
M3	0.30	0.10–0.92	3.35	0.92–12.24	0.37	0.19–0.70	6.06	3.44–10.66

GWG: gestational weight gain; pBMI: pregestational body mass index; M1: crude; M2: age-adjusted; M3: adjusted for age, socioeconomic status, education level, and gynecologic age; OR: odds ratio: CI: confidence interval.

**Table 4 nutrients-16-03726-t004:** Predictor variables of SGA newborn birth in teenager mothers in Colombia and Mexico.

	Colombia, *n* = 294		Mexico, *n* = 396	
	OR	95% CI	OR	95% CI
≤7 antenatal care visits				
M1	7.34	0.97–55.21	2.75	1.20–6.29
M2	7.05	0.93–53.19	2.60	1.13–5.97
M3	6.91	0.91–52.19	2.70	1.16–6.28
GWG insufficient				
M1	1.887	0.721–4.935	1.271	0.785–2.059
M2	1.889	0.717–4.975	1.209	0.741–1.972
M3	1.698	0.625–4.614	1.140	0.685–1.897
<15 years old				
M1	5.31	0.69–40.93	1.15	0.70–1.87
M2	7.89	0.89–64.33	1.10	0.67–1.81
M1	5.31	0.69–40.93	1.15	0.70–1.87
Lives with the baby’s father				
M1	1.09	0.42–2.46	0.84	0.51–1.38
M2	1.09	0.48–2.47	0.85	0.51–1.40
M3	1.04	0.45–2.41	0.85	0.51–1.40
Without partner support				
M1	1.26	0.56–2.85	1.57	0.63–1.91
M2	1.26	0.55–2.84	1.56	0.63–3.90
M3	1.31	0.57–3.02	1.55	0.62–3.89
Without parents support				
M1	1.48	0.65–3.37	0.96	0.48–1.93
M2	1.48	0.65–3.37	0.99	0.49–2.00
M3	1.37	0.59–3.17	0.98	0.48–1.99
Without extended family support				
M1	0.94	0.33–2.73	0.53	0.19–1.49
M2	0.91	0.32–2.71	0.49	0.17–1.40
M3	0.92	0.32–2.75	0.50	0.18–1.43
Antenatal care initiated after the second trimester				
M1	1.54	0.68–3.48	1.07	0.52–2.15
M2	1.56	0.68–3.55	1.07	0.53–2.16
M3	1.56	0.69–3.55	1.69	0.53–2.14
Menarche onset <11 years old				
M1	0.81	0.81–2.16	1.01	0.61–1.66
M2	0.89	0.33–2.38	1.05	0.64–1.73
M3	0.93	0.34–2.53	1.06	0.64–1.74

SGA: small for gestational age; M1 Crude, M2 adjusted for age; M3 adjusted for age, socioeconomic status, educational lag, pBMI, gynecologic age.

## Data Availability

To access data supporting the results of this study, please submit a formal request with a clear justification to the National Institute of Perinatology.

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
