# Peer review of "Low Antenatal Care Number of Consultations Is Associated with Gestational Weight Gain and Birth Weight of Offspring of Teenage Mothers: A Study Based on Colombian and Mexican Cohorts"

_nutrients, 2024, doi:10.3390/nu16213726_

Round 1
Reviewer 1 Report
Comments and Suggestions for Authors
Low antenatal care number of consultations is associated with gestational weight gain and birth weight of offspring of teen- age mothers: a study based on Colombian and Mexican cohorts.
PREAMBLE :
This is an interesting and serious study on teenage pregnancies (11-19) in Colombia and Mexico reporting 690 cases.
The conclusion of this study : « The study concludes that excessive and insufficient GWG, and SGA were associated with inadequate ANC consultation numbers. It is essential to promote antenatal care in adolescent pregnancy to avoid suboptimal GWG and SGA. » is of interest for public health and the future management of these teenage pregnancies.
Particularly, in Mexico, health workers must work to avoid in the future that 64% of their teenagers begin their antenatal care only during the second trimester (a sentence should emphasize this in the conclusion of the paper.)
This study brings very valuable socioeconomical informations in these 2 countries living with partner, extended families, financial support, educational lag, the fact that in Mexico there were 46% of girls < 15 years ( !), 64% of girls beginning the antenatal care in the second trimester, age at menarche etc….).
It is difficult for me to critic this study which obviously took hundred of hours and efforts to be achieved, with a very good methodology of analysis and exclusions for the final cohort. The difficulty comes from the fact that the authors, at reason, have used 2 international recommendations which are now more and more contested:
- Intergrowth21 for diagnosis of SGA
- IOM2009 recommendations for optimal gestational weight gain (GWG)
IOM2009 recommendations are adequate for normal and overweight women but absolutely wrong for, on one side, underweight women (too low) and, on the other, obese ones (too high). But it is not an insurmontable problem in this study as underweight women were 3-4% and obese 2-3% (Table 2).
Robillard PY, Dekker G, Boukerrou M, Le Moullec N, Hulsey TC. Relationship between pre-pregnancy maternal BMI and optimal weight gain in singleton pregnancies. Heliyon. 2018 May 10;4(5):e00615. doi: 10.1016/j.heliyon.2018.e00615.
NB : the authors are not obliged to cite this in their text, as in their cohort underweight and obese girls were negligeable.
Intergrowth21 recommendations are different as SGA is a fundamental outcome of the present study. Intergrowth21 to me is an estimable and perfectly homourable daydream of an integration of all mankind in a single curve (it would be the Holy Grail). In contrary, it is now more and more contested around all the world. Apparently, also, the following sentence suggests that it has to be contested in Colombia and Mexico as well :
Specific consequence for the present study : rates of SGA for Colombia 17% (!) and for Mexico 24% (!!) instead of the consensual around 10%.
WHAT TO DO :
1) In limitations precise something like: « Following the international recommendations in this study, we have used the Intergrowth21 curves and found 17% of SGA in Colombian and 24% in Mexican teenagers instead of 10%. Intergrowth21 may be inadequate for our population and is now contested by many reports all around the world coming from different ethnic populations….. »
You may cite some of these references :
Grantz KL, Hediger ML, Liu D, Buck Louis GM. Fetal growth standards: the NICHD fetal growth study approach in context with INTERGROWTH-21st and the World Health Organization Multicentre Growth Reference Study. Am J Obstet Gynecol. 2018 Feb;218(2S):S641-S655.e28. doi: 10.1016/j.ajog.2017.11.593. Epub 2017 Dec 22. PMID: 29275821; PMCID: PMC5807181.
Xiao WQ, Zhang LF, He JR, Shen SY, Funk AL, Lu JH, Wei XL, Yu J, Yang L, Li F, Xia HM, Qiu X. Comparison of the INTERGROWTH-21st standard and a new reference for head circumference at birth among newborns in Southern China. Pediatr Res. 2019 Oct;86(4):529-536. doi: 10.1038/s41390-019-0446-0. Epub 2019 Jun 3. PMID: 31158843.
Cheng Y, Leung TY, Lao T, Chan YM, Sahota DS. Impact of replacing Chinese ethnicity-specific fetal biometry charts with the INTERGROWTH-21(st) standard. BJOG. 2016 Sep;123 Suppl 3:48-55. doi: 10.1111/1471-0528.14008. PMID: 27627597.
Sletner L, Kiserud T, Vangen S, Nakstad B, Jenum AK. Effects of applying universal fetal growth standards in a Scandinavian multi-ethnic population. Acta Obstet Gynecol Scand. 2018 Feb;97(2):168-179. doi: 10.1111/aogs.13269. Epub 2017 Dec 20. PMID: 29192969.
Gleason JL, Reddy UM, Chen Z, Grobman WA, Wapner RJ, Steller JG, Simhan H, Scifres CM, Blue N, Parry S, Grantz KL. Comparing population-based fetal growth standards in a US cohort. Am J Obstet Gynecol. 2024 Sep;231(3):338.e1-338.e18. doi: 10.1016/j.ajog.2023.12.034. Epub 2023 Dec 25. PMID: 38151220; PMCID: PMC11196385.
Anderson NH, Sadler LC, McKinlay CJD, McCowan LME. INTERGROWTH-21st vs customized birthweight standards for identification of perinatal mortality and morbidity. Am J Obstet Gynecol. 2016 Apr;214(4):509.e1-509.e7. doi: 10.1016/j.ajog.2015.10.931. Epub 2015 Nov 4. PMID: 26546850.
2) The authors should replace everywhere in the text the word « adolescent » by « teenage ». Your study was including the ages 11-19 (it is even stated in the title). Adolescents mean the ages 11-17. There are these ambiguities everywhere in the text. Adding the ages of 18 and 19 at least doubles the number of girls being included in your cohort.
3) Cesarean section. The rates of C-section are abnormally high in this study : 22% in Colombia ( !) and 50% (!!!!!) in Mexico. THIS in the conclusion must be stated as a future urgent public health action notably in Mexico : lowering this pathological rate. This has inevitably a snowball effect for the following deliveries.
The authors honestly state page 13 : « Adolescent pregnancy has consistently been associated with a lower incidence of cesarean births in studies conducted across various regions of the world [7,61–63]. »
The authors may add :
Robillard PY, Hulsey TC, Boukerrou M, Bonsante F, Dekker G, Iacobelli S. Linear association between maternal age and need of medical interventions at delivery in primiparae: a cohort of 21,235 singleton births. J Matern Fetal Neonatal Med. 2018 Aug;31(15):2027-2035. doi: 10.1080/14767058.2017.1334049. Epub 2017 Jun 26. PMID: 28532289; PMCID: PMC5743766.
In this study, astonishingly 12-13 years old deliver better (rate of need of c-section) than than girls 15-16 for example, and so on with increasing ages.
This study has many interest for future targeting of teenage pregnancies: focus on the specific socioeconomic populations to help in priority, absolutely work on the fact that in these girls begin their antenatal care during the 1st trimeter, and begin an absolute war against this incredible rate of C-sections.
Author Response
Reviewer 1
- Thank you very much for your valuable observations, which have enriched our manuscript. We especially appreciate your suggestion to include in the conclusions the need for healthcare workers to intervene to reduce the percentage of adolescents who begin prenatal care during the second trimester (64%).
- Thank you for pointing out the use of two increasingly controversial international recommendations. We have reviewed the bibliographic references you suggested for both recommendations and have incorporated them into the manuscript. We have also considered your valuable observations, which have been included in the limitations section.
- Regarding the IOM-2009 recommendations, these are appropriate for women with normal and overweight weights, but erroneous for women with underweight (too low) and obese (too high) weights. However, this does not represent an insurmountable problem in this study, as underweight women represented 3-4% and obese women 2-3%.
- Concerning Intergrowth-21 for the diagnosis of SGA, while the international recommendations were used in this study, we found 17% SGA in Colombian adolescents and 24% in Mexican adolescents, instead of the expected 10%. This is considered a limitation of our study, as the recommendations may be inadequate for our population and are currently being questioned in numerous international reports from various ethnic populations.
- The suggestion to replace "Adolescents" with "Teenagers" throughout the manuscript has been accepted and implemented.
- As you observe, the cesarean rates in this study are abnormally high. Scientific evidence indicates that Latin America has the highest cesarean rates, constituting a serious public health problem in Mexico. This is exacerbated by the high risk associated with adolescent pregnancy. Adolescents are often referred to tertiary-level institutions, often academic ones, where resident physicians, especially during evening and night shifts, make decisions regarding cesarean sections. This may be motivated by learning objectives, avoidance of continuous labor monitoring, and fear of lawsuits due to obstetric complications. We agree on the need to mention in the conclusions the urgency of reducing cesarean rates in this group of women, given the cumulative effect this has on future deliveries.
Reviewer 2 Report
Comments and Suggestions for Authors
In their retrospective study, the Authors compared two obstetric population from developing countries (Mexico vs Colombia) with a special focus on adolescent pregnancy.
Methods are appropriate, as well as references. Results are interesting and in line with the expected. The paper's readibility is good.
In the opinion of this reviewer there are two aspects requiring additional efforts:
1. the explication to choose two similar countries and not a match between developed and developing country
2.to add info on `perinatal outcomes (i.e., maternal and neonatal complications, such as gestational diabetes, hypertension, NICU admission etc.)
Comments on the Quality of English LanguageGood
Author Response
Reviewer 2
- Explanation for choosing two similar countries instead of a combination of developed and developing countries.
Response: This study selected two developing countries with similar adolescent birth rates to avoid biases that could affect the relationship between gestational weight gain, low birth weight for gestational age (LBWGA), and the number and timing of initiation of prenatal care in pregnant adolescents.
- Include information on perinatal outcomes (i.e., maternal and neonatal complications such as gestational diabetes, hypertension, NICU admission, etc.).
Response: To obtain a homogeneous group of adolescents and avoid confounding variables such as gestational diabetes, preeclampsia, prematurity, and neonatal intensive care unit (NICU) admission, all adolescents with chronic or infectious diseases, or complications during pregnancy or the neonatal period were excluded.
Round 2
Reviewer 2 Report
Comments and Suggestions for Authors
The Authors have addressed the issues